



# Linear and Non-linear Stability Analysis of the Rate and State Friction Model with Three State Variables

Nitish Sinha, Arun K. Singh

Visvesvaraya National Institute of Technology, Nagpur-440010, INDIA

nitishme08@gmail.com          aksinghb@gmail.com

**Abstract**

In this article, we study linear and non-linear stability of the three state variables rate and state friction (3sRSF) model with spring-mass sliding system. Linear stability analysis shows that critical stiffness, at which dynamical behaviour of the sliding system changes, increases with number of state variables. The bifurcation diagram reveals that route of chaos is period doubling and this has also been confirmed with the Poincaré maps. The present system is hyperchaos since all Lyapunov exponents are positive. It is also established that the 3sRSF model is more chaotic than corresponding to the 2sRSF model. Finally, the implication of the present study is also discussed.

## 1. Introduction

One of the most important applications of friction in recent decades is in understanding the sliding dynamics of earthquake faults (Brace and Byerlee, 1966; Dieterich, 1979; Rice and Ruina, 1983). It is believed that the stick-slip process along the earthquake faults results in earthquakes. Researchers use rate and state friction(RSF) model oftenly to explain the earthquake process (Brace and Byerlee, 1966; Dieterich, 1979; Rice and Ruina, 1983). The RSF model was proposed by Dieterich (1979,1981), Ruina (1983) and Ruina and Rice (1983). Although the RSF model is an empirical model, its genesis has been explained using the Eyering's rate reaction theory (Rice et. al., 2001). Classical Amontons-Coulombs' (AC) laws are widely used for explaining variety of friction based phenomena of hard solids (Persson,2000). Nonetheless these friction laws do not explain  many observations for instance increase in friction with time of contact and sliding velocity, more significantly, stiffness dependence of stick-slip behavior etc. (Rice and Ruina,1983). In fact, these



limitations of the AC laws led to the proposal of the modified friction model which is known
as the rate and state friction (RSF) model. According to this friction model of hard solids such
as rock solids depends on the "slip rate" as well as the "state" of the sliding surfaces (Rice and
Ruina,1983; Ruina,1983). Although one state variable explains well the stiffness dependence
of stick-slip oscillatory motion of a sliding mass, it doesn't explain its chaotic behavior. As a
result, one state variable RSF law has been modified by introducing an additional state
variable by believeing that chaos is a manifestation of more complex friction processes at the
slip interface. This observation led to the proposal of the two state variables rate and state
dependent friction (2sRSF) model. The 2sRSF model shows the chaotic behavior ( Ruina,
1983; Gu et. al., 1984; Gu and Wong, 1994; Zhiern and Dangmin, 1994; Niu and Chen, 1995;
Becker, 2000; Gao, 2013). It arises naturally a question what happens to the 2sRSF model if
one more state variable is added in this friction model. In this article we have studied
numerically linear and nonlinear dynamics of the three state variables rate and state
friction(3sRSF) with spring-mass sliding system. The results are also compared with the
corresponding two state variables rate and state friction (2sRSF) model.
Chaos is defined as "Aperiodic long-term behavior in a deterministic system that exhibits
sensitive dependence on initial conditions" (Strogatz,1994). The conditions for a continuous
dynamical system to be chaotic are that the governing differential equation must possess at
least three independent variables, and also show the dependence on initial conditions
(Devany,1989). There are many well known and extensively studied chaotic systems in
literature for example Duffing oscillator, Lorenz system, Rössler system etc. (Strogatz,1994).
Moreover, phase plot, Poincaré maps, bifurcation diagram, Lyapunov exponents etc. are the
numerical tools which are widely used for studying chaotic behavior of a dynamical system.
Rössler introduced the concept of hyperchaos by modifying one of the simplest chaotic
models (Rössler,1979). The general conditions for the hyper-chaos are that the system of

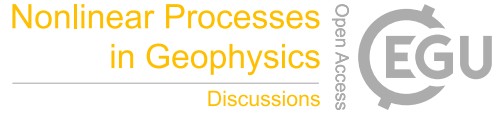

differential equations should have at least four independent variables and  the system must
also be dissipative (Wang and Wang,2008; Chen et. al., 2006). Moreover, the Lyapunov
exponents of the dynamical system must show at least two positive, one zero and one negative
(Niu and Chen,1995). Further, the sum of all Lyapunov exponents must be negative
(Moghtadaei and Goplaegani,2012). In additions to these conditions, the phase plot should
also show twisting structure in the chaotic behavior (Moghtadaei and Goplaegani, 2012).
Notwithstanding the aforementioned conditions for hyperchaos, there are dynamical systems
which have been claimed to be hyper chaos. For example, Oteski et al. (2015) have claimed
that  an air-filled differentially heated cavity to be hyperchaotic on the basis of  all positive
Lyapunov exponents(LEs). In the present 3sRSF model as well, we will establish numerically
that all LEs are  positive hence the 3sRSF dynamical system to be hyperchaotic.
In literature majority of study has been done with one state variable based RSF law (Ranjith
and Rice, 1999). The reason may be attributed to the fact that one state variable based friction
law is enough to explain the stick-slip phenomenon or frictional instability of hard surfaces.
Gu et al. (1984) have studied numerically the linear and non-linear behaviour of the spring-
mass slider with the 1sRSF model  as well as the 2sRSF model. They have reported stick-slip
behavior with 1sRSF model while the 2sRSF model shows  the period doubling as well as
chaotic behaviour. Gu and Wong (1992) have carried out  linear and nonlinear stability
analysis with both the1sRSF and 2sRSF models using the tools phase portraits, time series,
and bifurcation diagrams. They have established that the most significant parameter is  spring
stiffness  which controls the stability of the sliding mass. Zhiren and Dangmin (1994,1995)
have carried out the numerical simulations of 2sRSF model with the  slip law, and they
observed that the sliding system shows the quasi-periodic to chaotic behaviour upon decrease
in spring stiffness even in the absence of inertia that is, under the quasistatic conditions. They
have also estimated the Lyapunov exponents as well as Lyapunov dimensions to confirm the



evidence of chaotic behaviour of the system (Niu and Chen, 1995). Xuejun(2013) has
investigated the stability of the 2sRSF and finds the period doubling route to chaos. Wang
(2002,2009) has pointed out that the "slip" and "slowness" laws differ in high velocity
regime but not in the low velocity sliding regime. In recent times the 2sRSF model has been
used to validate the experimental data concerning rock friction at high temperature in the
framework of the 2sRSF(Liu, 2007 King and Marone,2012). Nontheless these researchers
have not reported any evidence of chaotic behavior in the experiments at high temperature.
The present analysis is related with the three state variable RSF model i.e., the 3sRSF model.
The organization of the paper is as following. First we have derived governing differential
equations of the spring-mass sliding system with 3sRSF in non-dimensional form following
the same procedure as was done by Xuejun[2013]. It is then linear stability of Eq. (4) is
carried out by linearizing about steady state or equilibrium points. The expression for critical
stiffness is also derived using Routh- Hurwitz criterion (Persson,2000). The physical meaning
of the critical stiffness is that at this value of stiffness the sliding behavior changes from
unstable to stable sliding or vice versa. The non-linear analysis of Eq. (4) is also carried out in
detail with different tools such as phase plot, Poincaré maps, bifurcation diagram, Lyapunov
exponents and Lyapunov dimensions. Finally a comparative study is also done between
2sRSF and 3sRSF models to justify the present results.
**2.Modelling of Spring-mass system with three state variables friction law**
According to the rate and state friction(RSF) model, frictional stress '$\tau$' of a sliding hard
surface depends on sliding velocity '$v$' and state variable '$\theta$' (Ruina, 1983). Based on the
experimental observations Dieterich(1978), Ruina(1980,1983), Ruina and Rice(1983)
proposed the following empirical relation
$$\tau = \tau^* + \theta_i + A \ln \frac{v}{v^*} \text{ , and } \frac{d\theta_i}{dt} = -\frac{v}{L_i}\left[\theta_i + B_i \ln \frac{v}{v^*}\right]. \tag{1}$$





where $\theta_i$ is number $(i=1,2,3...)$ of state variables, $B_i$ are $L_i$ are the constants. Further '$\tau^*$' and
'$v^*$' are reference frictional shear stress and shear velocity respectively. The system of
differential equations in Eq.(1) with three state variables are expanded as

$$\tau = \tau^* + \theta_1 + \theta_2 + \theta_3 + A\ln\frac{v}{v^*}, \& \frac{d\theta_1}{dt} = -\frac{v}{L_1}\left[\theta_1 + B_1\ln\frac{v}{v^*}\right].$$

$$\frac{d\theta_2}{dt} = -\frac{v}{L_2}\left[\theta_2 + B_2\ln\frac{v}{v^*}\right], \& \frac{d\theta_3}{dt} = -\frac{v}{L_3}\left[\theta_3 + B_3\ln\frac{v}{v^*}\right]. \tag{2}$$

Dieterich(1979), Ruina(1983) have proposed two laws governing the "state" of the sliding
surfaces which are know as the Ruina-Rice slip law or simply slip law and Dieterich-Ruina
ageing law or ageing law [3]. It is important to note that the, unlike ageing law, the slip law of
the RSF model shows chaotic behaviour (King and Marone,2012 ). The reason for this
contradictory observation is not yet reported in literature.
In order to study the 3sRSF model, we have also used the spring-mass sliding system under
the quasi-static conditions. The free end of the spring having spring constant $k(Pam^{-1})$ is
being pulled constantly with a constant pulling velocity '$v_0$' as a result the rate of change of
friction at the sliding interface is given by

$$\frac{d\tau}{dt} = k(v_0 - v). \tag{3}$$

The non-dimension form of above set of Eqs.(1-5) are expressed by introducing non-
dimension variables as velocity $\phi$, shear stress $f$, state variables $\hat{\theta}_1$, and $\hat{\theta}_2$, time T, pulling
velocity $\phi_0$ ,spring stiffness K

$$f = \frac{\tau - \tau^*}{A}, \quad \phi = \ln\frac{v}{v^*}, \quad \hat{\theta}_1 = \frac{\theta_1}{A}, \quad \hat{\theta}_2 = \frac{\theta_2}{A}, \quad T = \frac{v^*}{L_1}t, \quad \beta_1 = \frac{B_1}{A}, \quad \beta_2 = \frac{B_2}{A}, \quad \beta_3 = \frac{B_3}{A}, \quad \rho = \frac{L_1}{L_2},$$

$$\rho_1 = \frac{L_1}{L_3}, \quad \phi_0 = \ln\frac{v^0}{v^*}, \quad K = k\frac{L_1}{A}.$$



Non-dimensional form of the system of differential equations Eq.(4) is obtained using Eqs. (2-
3). After having eliminated the third state variable $\hat{\theta}_3$, we get the following system of
differential equations:

$$
\begin{cases}
\dfrac{d\phi}{dT} = e^{\phi}\left[(1-\rho_1)\hat{\theta}_1 + (\rho-\rho_1)\hat{\theta}_2 + (\beta_1 + \rho\beta_2 + \rho_1\beta_3 - \rho)\phi + \rho_1 f - K\right] + Ke^{\phi_0} \\[2mm]
\dfrac{df}{dT} = K\left(e^{\phi_0} - e^{\phi}\right) \\[2mm]
\dfrac{d\hat{\theta}_1}{dT} = -e^{\phi}\left(\hat{\theta}_1 + \beta_1\phi\right) \\[2mm]
\dfrac{d\hat{\theta}_2}{dT} = -\rho e^{\phi}\left(\hat{\theta}_2 + \beta_2\phi\right)
\end{cases}
\tag{4}
$$

It may be noted that Eq.(4) is having four variables i.e, four dimensional system, thus we
investigate the possibility of hyperchaos in Eq.(4).
**3.0.Results and Discussion**
**3.1.Linear Stability analysis**
Linear stability of the spring-mass model is done about steady state or equilibrium
point(Strogratz, 1994). The equilibrium or fixed points are obtained by equating the equations
to zero. The equilibrium points of Eq.(4) are obtained as
$\phi_{ss} = \phi_0, \; \theta_{1ss} = -\beta_1\phi_0, \; \hat{\theta}_{2ss} = -\beta_2\phi, \; \text{and} \; f_{ss} = \left(\dfrac{\rho}{\rho_1} - \beta_1 - \beta_2 - \beta_3\right)\phi_0$    (5)
The characterstic equation $|J_0 - \lambda I| = 0$, is expanded for polynomial equation in terms of eigen
value $\lambda$. where $J_0$ is Jacobian matrix of Eq.(4) about the steady state and $I$ is identity matrix.
Routh-Hurwitz criterion is used to obtain critical stiffness $k_{cr}$ at which sliding behaviour of
the spring-mass system changes. Other details about evaluating $k_{cr}$ is given in appendix-I.
The physical significance of $k_{cr}$ is that the sliding system changes its behaviour from unstable
to stable sliding for spring stiffness larger than $k_{cr}$ (Gu et. al., 1984; Ranjith and Rice,1999).
For instance, Fig.1 presents the results that the sliding system is dynamically unstable for



stiffness $k = 0.2633$ and neutral for critical stiffness $k_{cr} = 0.2635$ and stable for stiffness
$k_{cr} = 0.2638$. The value of critical stiffness is evaluated numerically using the expression for
critical stiffness given in Appendix-I. Noting that the results in Fig.1 are in confirmation with
the 1s RSF model (Ranjith and Rice,1999).

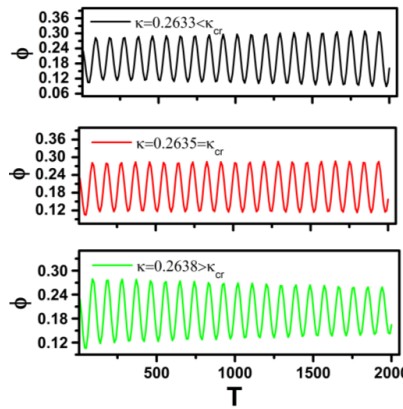

Fig.1. Stiffness dependent sliding behavior of spring-mass for $\beta_1 = 1.2$, $\beta_2 = 0.84$, $\beta_3 = 0.38$,
$\rho = 0.048$ and $\rho_1 = 0.034$ for initial condition [0.19885,-1.00824,-0.23862,-0.167034].
**3.2.Effect of friction parameters on critical stiffness**
The effect of friction parameter such as $\beta_1$ is investigated on critical stiffness $k_{cr}$ numerically.
The values of friction parameters are considerd the same as in literature[Rice and Ruina,1983;
Gu. et. al.,1984]. However numerical values of additional parameters $\beta_3$ and $\rho_1$ in the 3sRSF
model are estimated on the basis of the reported values in literature (Gu. et. al., 1984) For
instance, friction parameter $\beta$ decreases if friction law is modified from one state variable to
two state variables. The result in Fig.2 shows that $k_{cr}$ increases linearly with $\beta_1$. This linear
behaviour is also seen with the 2sRSF law though we are not presenting the results here. The
dependence of critical stiffness in the 3sRSF model with respect to variables, for instance
$\beta_2$, $\beta_3$, $\rho$ and $\rho_1$, is also linear though we have not presented the results here.

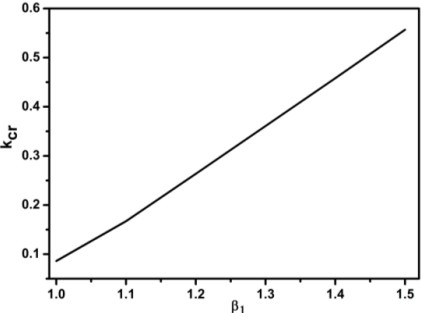

Fig.2. Effect of friction parameter $\beta_1$ on critical stiffness $k_{cr}$ for $\beta_2 = 0.84$, $\beta_3 = 0.38$,

$\rho = 0.048$ and $\rho_1 = 0.034$.

### 3.3.Nonlinear stability analysis

Motivated from linear stability analysis, we have also carried out non-linear stability of the

system of governing differential equations in Eq.(4). This is solved with MATLAB® using

*ode23s* solver for ordinary differential equations. Fig.3 shows the single orbit in phase

portrait, which means the system behaviour is periodic at spring stiffness $k = 0.087$. This has

also been confirmed using Poincaré section which shows single point in the map in Fig.3.

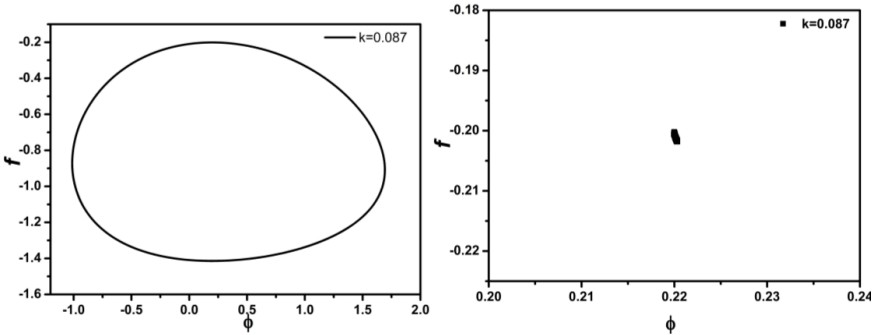

Fig.3 phase diagram(left) $f$ vs. $\phi$ and Poincaré section(right) for $k = 0.087$, $\beta_1 = 1.0$,

$\beta_2 = 0.84$, $\beta_3 = 0.38$, $\rho = 0.048$ and $\rho_1 = 0.034$ for initial condition [0,0,0,0].

Now upon lowering the magnitude of spring stiffness to $k = 0.085$, Fig.4 shows the evidence

of period doubling and this phenomena is also confirm by the Poincaré map. As Poincaré

section in Fig.4 shows two points.



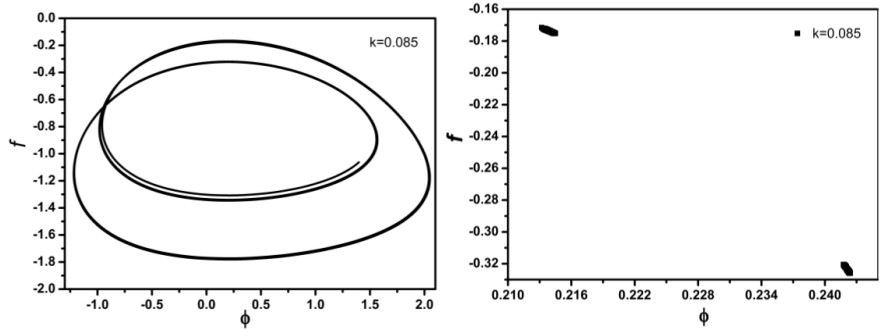

Fig.4 phase diagram $f$ vs. $\phi$ and corresponding Poincaré section for $k = 0.085$, $\beta_1 = 1.0$,
$\beta_2 = 0.84$, $\beta_3 = 0.38$, $\rho = 0.048$ and $\rho_1 = 0.034$ for initial condition [0,0,0,0].
As magnitude of stiffness decreases further to $k = 0.08437$, the dynamical behaviour of the
system changes further. Now the phase portrait in Fig.5 results in period quadrupling and this
is also confirmed in the corresponding Poincaré section in Fig.5.

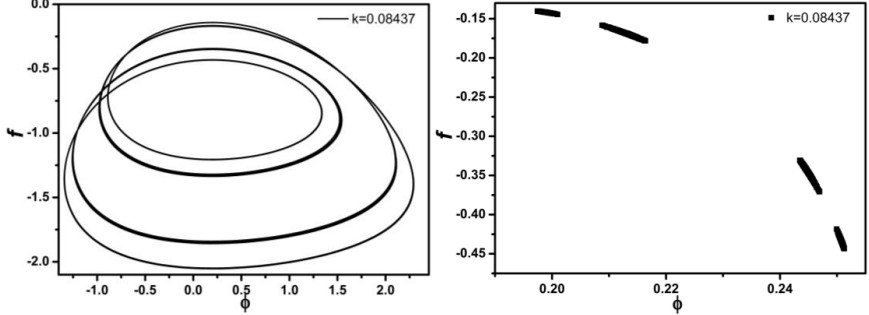

Fig.5 phase diagram $f$ vs. $\phi$ and corresponding Poincaré section for $k = 0.08437$, $\beta_1 = 1.0$,
$\beta_2 = 0.84$, $\beta_3 = 0.38$, $\rho = 0.048$ and $\rho_1 = 0.034$ for initial condition [0,0,0,0].
As the controlling parameter $k$ decreases further, the 3sRSF leads the spring-mass system in
chaos. For instance,Fig.6 presents the phase diagram and corresponding Poincaré section for
$k = 0.08421$. The phase portrait shows infinite period with bounded orbits and the
corresponding Poincaré section in the form of continuous line.



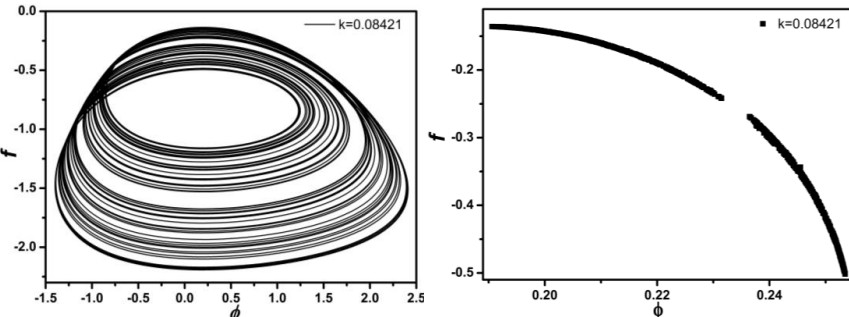

Fig.6   phase diagram $f$ vs. $\phi$ and corresponding Poincaré section for $k = 0.08421$, $\beta_1 = 1.0$,

$\beta_2 = 0.84$, $\beta_3 = 0.38$, $\rho = 0.048$ and $\rho_1 = 0.034$ for initial condition [0,0,0,0].

A physical significance of the present results is in nucleation of earthquake process. For

instance, The phase portraits in Fig.3-6 show an interesting observation  that  frictional stress

as well as corresponding slip velocity at the sliding interface changes from periodic  to chaotic

upon decreasing spring stiffness of the slider. This results in a direct surge of stress amplitude

thus the nucleation of earthquake process occurs.  This observation is similar to the chaotic

nature of the sliding mass with the 2sRSF in which magnitude of the stress fluctuates

considerably thus the earthquake nucleation begins (Becker, 2000).

**3.4 Bifurcation diagram**

The results in Figs.(3-6) have also been confirmed by the bifurcation diagram in shown Fig.7.

In the bifurcation diagram the control parameter in the form of non-dimensional stiffness $k$

decreases by a small step $10^{-6}$ from $k = 0.089$ to $k = 0.084$, the  evolution of the system is

initially periodic oscillation with increasing amplitude as evident in Fig.7. Upon further

decrease in stiffness upto $k = 0.085$, the behaviour of the system changes to period doubling

as obvious in Fig.7. If stiffness decreases to further lower value i.e., $k = 0.08437$, the system

behaviour bifurcates to the period four (Fig.7). Finally the system results in chaotic behaviour

at minimum stiffness $k = 0.08421$. These results are in confirmation with phase portraits and

Poincaré section in Figs.(3-6).



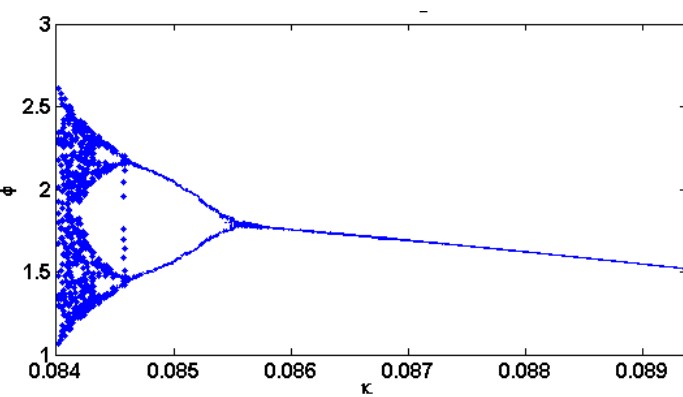

Fig.7 Bifurcation diagram for spring stiffness $k = 0.87$ to $0.084$, $\beta_1 = 1.0$, $\beta_2 = 0.84$,
$\beta_3 = 0.38$, $\rho = 0.048$ and $\rho_1 = 0.034$ for initial condition [0,0,0,0].
Fig.7 also summarizes the variation of velocity amplitude with decreasing spring stiffness. It
is obvious that the overall velocity amplitude increase from periodic to chaotic way as
stiffness of the connecting spring decreases. This observation is consistent with the phase
plots in Figs.3-6.
**3.5 Lyapunov exponent and dimensions**
Lyapunov exponent(LE) is the most significant tool for investigating the dynamical behavior
of a physical system (Kaplan and Yorke,1979). We have used the MATLAB®program for
evaluating the LE of the present dynamical system by Lyapunov Exponent Toolbox (LET),
which is developed by Steve SIU (1998). For the four-dimensional dissipative system, there
are three possible type of strange attractors such as the combination of Lyapunov spectra as
(+,+,0,-), (+,0,0,-) and (+,0,-,-) (Wolf. et. al., 1985). If LE is negative the dynamical system is
stable with dissipative in nature, while the positive LE signifies the system become unstable
orbit or chaotic. However, LE with zero magnitude signifies the system is dynamically neutral
(Wolf. et. al., 1985).
The present analysis of the 3sRSF model shows in Fig.8 that the magnitude of LEs are
$LE_1 = 1.8146$, $LE_2 = 0.0461$, $LE_3 = 0.0577$, $LE_4 = 0.0351$. This result confirms that the present
dynamical system is very similar to a hyperchao as more than one Lyapunov exponents is



positive (Oteski. et. al., 2015). At the same time, the magnitude of three LEs are one order
less than the remaining one. This result is in contrast with the 2sRSF in which results are one
positive, one negative and one zero in magnitude (Niu and Chen,1995). The relationship
between the Lyapunov exponents and fractal dimensions is established by Kaplan and
Yorke(1979). They have proposed the Lyapunov or Kaplan-Yorke dimension $D_{KY}$ which is
given by the formula: $D_{KY} = D + \dfrac{1}{|h_D + 1|} \sum\limits_{i=1}^{D} h_i$
where $D$ is the largest integer for which $\sum\limits_{i=1}^{D} h_i > 0$. As a result, $D_{KY}$ is a convenient
geometrical measure of objects in phase space if Lyapunov exponents are known. The fractal
dimension of the present dynamical system is calculated to be as 5.70.

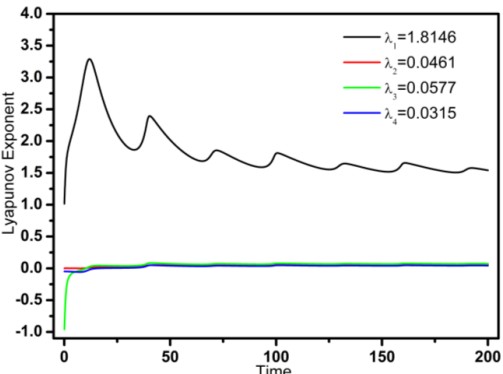

Fig.8.Lypunov exponents vs. .time for k=0.08421, $\beta_1 = 1.0$, $\beta_2 = 0.84$, $\beta_3 = 0.38$, $\rho = 0.048$
and $\rho_1 = 0.034$ for initial conditions [0,0,0,0]
The 3sRSF based quasistatic system also follows the universal period doubling route to chaos.
The Feigenbaum number is estimated using given formula $\delta_n = (k_{n+1} - k_n)/(k_{n+2} - k_{n+1})$ where
$n = 1, 2, 3...$, this number should converge to Feigenbaum number 4.669201. we have
calculated Feigenbaum universality constant for 3sRSF law and estimated to 3.9375. However
this single value does not indicate the sign of convergence. It may be possible that



convergence for the bifurcation sequence to chaos for the friction model is different from the
logistic map.
We have also investigated whether the present friction model fulfils the other conditions of
hyperchaos. The hyperchaotic behavior in the form of phase portraits in Fig.9 shows the
twisting nature of the phase diagram. This is also a feature of hyper chaos().

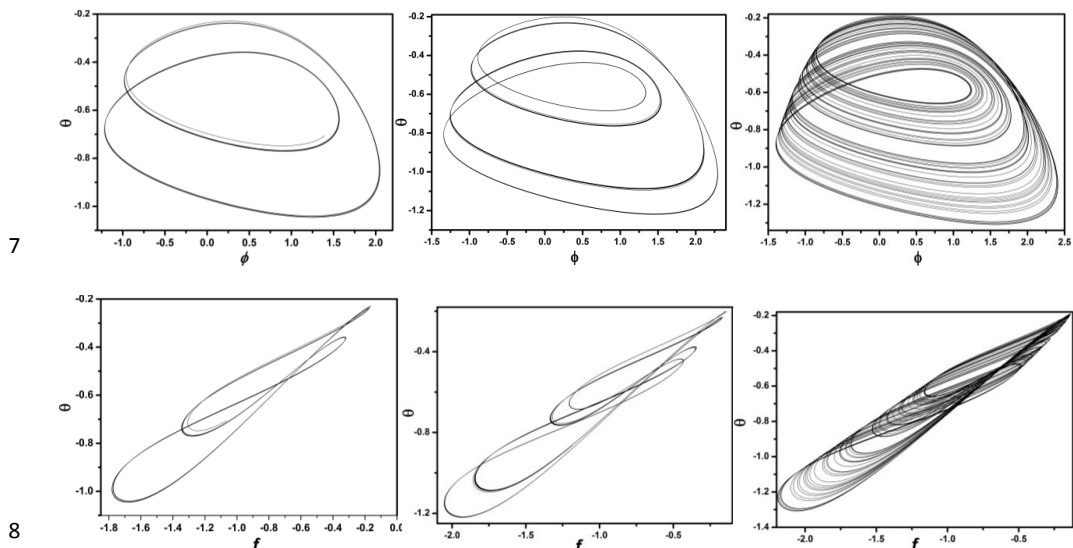

Fig.9. Twisted phase diagram for stiffness value (a) $k = 0.085$, (b) $k = 0.08437$, (c)
$k = 0.08421$ and $\beta_1 = 1.0, \beta_2 = 0.84, \beta_3 = 0.38, \rho = 0.048$ and $\rho_1 = 0.034$ for initial condition
[0,0,0,0].
We have also compared the linear and non-linear behavior between the 2sRSF and the 3sRSF
models. For instance, critical stiffness, at which dynamics of stick-slip motion changes,
increases with number of state variables. Moreover,  the route of chaos is same for both
2sRSF and 3sRSF models, that is, period doubling. But  period eight and period sixteen are
not observed in the present system which is unlike to the 2sRSF model (Xuejun,2013).
Moreover,LEs of the 2sRSF are reported to be one positive, one negative and one  zero.  The
3sRSF model, in contrast, shows all four LEs are positive. This result has been confirmed




using magnitude of fractal dimension (FD). For example, FD of the 2sRSF is 2.11 which is

less than the FD of 3sRSF i.e., 5.7. Moreover, Poincaré section of the 3sRSF model is

slightly intricate than the 2sRSF model. On the basis of these evidances, it is established that

the 3sRSF model is more chaotic than the 2sRSF model.

**5.Conclusions**

We have established numerically that the three state variables based RSF model show the

chaotic behavior. All Lyapunov exponent is positive. The route of chaos is established to be

period doubling bifurcation. Moreover, critical stiffness of the dynamical system increases

with number of state variables. It is also observed that the 3sRSF is more chaotic than

corresponding to the 2sRSF. It is shown that the 3sRSF model is hyperchaotic as it exhibits

all positive Lyapunov exponents.

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

**Appendix -I**
It  Jacobian matrix corresponding to the equilibrium point may be expressed as

$$J_0 = \begin{bmatrix} (\beta_1 + \rho\beta_2 + \rho_1\beta_3 - \rho)e^{\phi_0}, & \rho e^{\phi_0}, & (1-\rho_1)e^{\phi_0}, & (\rho - \rho_1)e^{\phi_0} \\ -Ke^{\phi_0}, & 0, & 0, & 0 \\ -\beta_1 e^{\phi_0}, & 0, & -e^{\phi_0}, & 0 \\ -\rho\beta_2 e^{\phi_0}, & 0, & 0, & -\rho e^{\phi_0} \end{bmatrix}_{(\phi_{ss}, \hat{\theta}_{1ss}, \hat{\theta}_{2ss}, f_{ss})}$$

The polynomial equation containing the eigen values in term of $\lambda$ is obtained from the
expansion of the above Jacobian matrix  as following

$$\lambda^4 + e^{\phi_0}\left(1 + \rho + \rho_1 + K - \beta_1 - \rho\beta_2 - \rho\beta_3\right)\lambda^3 +$$
$$e^{2\phi_0}\left(K + \rho + \rho_1 + \rho\rho_1 + 2K\rho - \rho\beta_1 - \rho\beta_2 - \rho_1\beta_1 - \rho\beta_3 - \rho\rho_1\beta_2 - \rho\rho_1\beta_3\right)\lambda^2$$
$$+ e^{3\phi_0}\left(2K\rho + \rho\rho_1 + \rho\rho_1\beta_1 - \rho\rho_1\beta_2 - \rho\rho_1\beta_3 + K\rho^2\right)\lambda + e^{4\phi_0}K\rho^2 = 0$$

$$s_0\lambda^4 + s_1\lambda^3 + s_2\lambda^2 + s_1\lambda + s_4 = 0$$
where:
$$s_0 = 1$$
$$s_1 = e^{\phi_0}\left(1 + \rho + \rho_1 + K - \beta_1 - \rho\beta_2 - \rho\beta_3\right)$$
$$s_2 = e^{2\phi_0}\left(K + \rho + \rho_1 + \rho\rho_1 + 2K\rho - \rho\beta_1 - \rho\beta_2 - \rho_1\beta_1 - \rho\beta_3 - \rho\rho_1\beta_2 - \rho\rho_1\beta_3\right)$$
$$s_3 = e^{3\phi_0}\left(2K\rho + \rho\rho_1 + \rho\rho_1\beta_1 - \rho\rho_1\beta_2 - \rho\rho_1\beta_3 + K\rho^2\right)$$
$$s_4 = e^{4\phi_0}K\rho^2$$

The Routh-Hurwitz criterion is used to get  the condition for stability of the present friction
model. The characteristic polynomial equation is obtained as $s_0\lambda^4 + s_1\lambda^3 + s_2\lambda^2 + s_3\lambda + s_4 = 0$.
After applying the Routh-Hurwitz criteria $s_1 s_2 - s_0 s_3 = 0$ and $s_1 s_2 s_3 - s_0 s_3^2 - s_4 s_1^2 = 0$. These
non-linear algebraic equations are in turn, solved numerically for evaluating critical stiffness.