# Peer review of "Linear and Non-linear Stability Analysis of the Rate and State Friction"

_Nonlinear Processes in Geophysics, 2016_

## Referee Comment (RC1) · Anonymous Referee #1 · 29 Feb 2016

This paper begins by looking at a class of rate and state friction models for spring-mass systems developed in the 1980s. Basically, it is new in that it deals with 3 state variables in this application. Turcotte among others has shown that in a number of papers that slider-block models can be related to lattice models. Moreover, Gabrielov et al. (1994) Phys. Rev E. 50, 188-197 found 3 element configurations calculating Poincare sections, etc. just as in this new paper. So, this kind of configuration is not really new. The question that emerges is WHY the need to go to such a system; simply saying that 3 degrees of freedom is necessary for chaos is superficial. Are there any PHYSICAL features of the underlying problem that the new models (including their 2 state model) describe that the original 1980s models cannot. There is no connection with the under-

lying physics presented here, and the authors simply use some canned programs to calculate Lyapunov exponents (why do they get more than one; that does not conform with the traditional definition for the Lyapunov exponent) and obtain a fractal dimension of 5.7 (why, when dealing with a 3 degree of freedom system). Indeed, it is not sufficient for them to apply these tools to data generated from their new model without asking what kind of outcomes emerge from the original 1980s models. Furthermore, does this model given any added physical insight into the frictional problem that motivated the original studies? Just because their new model is more complicated does not of itself justify its publication unless compelling reasons emerging from the physical problem of frictional slip etc. as well as a clear identification of the meaning of their MatLab results is provided.

---

## Author Comment (AC1) · 6 Mar 2016

Dear Referee,

Thanks for your constructive comments. We have tried our best to rebuttal your comments line by line as following:

**(1)Referee: WHY the need to go to such a system; simply saying that 3 degrees of freedom is necessary for chaos is superficial.**

Authors: Based on the experimental observations, Ruina (1980, 1983), Ruina and Rice (1983) have proposed the following empirical relation for rock friction

$$\tau = \tau^* + A \ln\left(v/v^*\right) + B \ln\left(v^*\theta_i/L_i\right).$$

The state variables '$\theta_i$' are defined using the slip law of friction as

$$\frac{d\theta_i}{dt} = -\frac{v\theta_i}{L_i} \ln\left(\frac{v\theta_i}{L_i}\right). \text{ where } i = 1,2,3$$

The above model with one state variable based RSF model(1sRSF) is generally used to explain stick-slip behaviour but this model doesn't explain the chaotic behaviour of the sliding system. However, the RSF law with two state variables i.e., 2sRSF does show the chaos (Gu et al., 1984; Gu and Wong, 1994; Niu and Chen, 1994, 1994; Becker, 2000). As a result, a natural question arises how the non linear behaviour of the RSF model changes if one more state variable is added in the friction model that is, the RSF model with three state variables (3sRSF). This is what we have analyzed in this paper without paying much attention on the practical/physical significance of the 3sRSF model. In recent times, the RSF model with two state variables (2sRSF) are widely being used to validate the sliding friction of rocks at higher temperatures by justifying the multiple mechanisms of friction becomes active in such a scenario (King and Marone, 2012; Lui, 2007). We believe that the 3sRSF model could be promising in high sliding and temperature experiments on rock surfaces.

**(2)Referee: Are there any PHYSICAL features of the underlying problem that the new models (including their 2 state models) describe that the original 1980s models cannot.**

Authors: It is to be mentioned that the original RSF model (1980s) was proposed with one state variable (1sRSF) model. However Ruina (1980, 1983) argued that 1sRSF model is not sufficient to explain friction experiments on rock surfaces. He used the 2sRSF model to fit the experimental data and also justified the need of 2sRSF. Gu et al. (1984) studied the 2sRSF law numerically and predicted its chaotic behaviour. In recent times, King and

Marone ( 2012) have reported that the 2sRSF explains better the experimental data pertaining to high temperature sliding than the1sRSF model. They have attributed the reason to the possible onset of a second mechanism of friction at higher temperatures.

**(3)Referee: There is no connection with the underlying physics presented here, and the authors simply use some canned programs to calculate Lyapunov exponents (why do they get more than one; that does not conform with the traditional definition for the Lyapunov exponent) and obtain a fractal dimension of 5.7 (why, when dealing with a 3 degree of freedom system).**

Authors: Addition of one more state variable in the 2sRSF is believed to explain more complex form of friction at rock surfaces, for instance at high temperature and sliding velocities. Moreover, the number of Lyapunov exponents (LEs) in a dynamical system is generally equal to the number of the degree of freedom (dimensions) of the system. But the largest value of the LE is generally reported. As Niu and Chen (1994), Becker (2000) have reported all the Lyapunov exponents in the published paper. Following them, we have also presented all the Lyapunov exponents in the present paper. Moreover, calculation of the fractal dimension of the present dynamical system, which is equal to 5.7, is not correct at all. Since all the Lyapunov exponents are positive thus the formula for calculating fractal dimension is not valid. As an error, we shall remove this estimation from the present article.

**(4)Referee: it is not sufficient for them to apply these tools to data generated from their new model without asking what kind of outcomes emerge from the original 1980s models.**

Authors: The original 1980s RSF models explained well stiffness dependence of stick-slip motion of hard surfaces. The original 1980s RSF models also include two state variables (2sRSF) but the chaotic behaviour of the 2sRSF was not studied in detail in those times. In 1990s, Gu and Wong (1994), Niu and Chen (1994,1995), Becker (2000) studied the non-linear behaviour of the 2sRSF model in detail using non-linear dynamical tools such as Poincare map, Bifurcation diagram, Lyapunov exponents etc. These studies firmly established the route of chaos as "Period doubling". Further, Niu and Chen (1995), Becker(2000) reported the fractal dimension of 2sRSF is equal to 2.11, and have also reported the Feigenbaum number which nearly converges to universal Feigenbaum constant (4.669201).

**(5)Referee: does this model given any added physical insight into the frictional problem that motivated the original studies?**

Authors: Yes, the present friction model has revealed that all the Lyapunov exponents are positive. This observation is, in contrast, with the 2sRSF model which shows one positive, one negative and one nearly equal to zero (Niu and Chen, 1995, Becker, 2000). Moreover, despite being the same route of chaos that is "period doubling", the 3sRSF system shows the chaotic behaviour after four periods while the 2sRSF results in chaos after sixteen periods. The reason is obvious due to number of positive LEs in these two systems. Further linear stability analysis shows that the critical stiffness of the sliding system predicted by the 3sRSF model is larger than the corresponding 2sRSF model. Accordingly, stiffness at which chaos occurs in the 3sRSF is also larger than the 2sRSF law ( Niu and Chen,1994). Further, stress drop during the chaos with the 3sRSF model is larger than the corresponding 2sRSF model. On the basis of these observations, we establish that the 3sRSF is more chaotic than the 2sRSF.

**(6)Referee: Just because their new model is more complicated does not of itself justify its publication unless compelling reasons emerging from the Physical problem of frictional slip etc. as well as a clear clarification of the meaning of their MatLab results is provided.**

Authors: Recent friction sliding experiments at high temperature and  sliding velocities have shown more complex form of friction. So the more complex form of the RSF model could be justified. Of course, further study needed to justify the addition of third state variables in the RSF model. Moreover, the physical meaning of MATLAB plots are explained as

[Figure]

Fig.3-6: Phase diagram $f$ vs. $\phi$ for $k = 0.087$, $k = 0.085$ , $k = 0.08437$ and $k = 0.08421$
$\beta_1 = 1.0$, $\beta_2 = 0.84$, $\beta_3 = 0.38$, $\rho = 0.048$ and $\rho_1 = 0.034$ for initial condition [0,0,0,0].

The plots in Figs.3-6 of the paper show the periodic behaviour with fixed oscillating amplitude of the stress level for spring stiffness $k = 0.087$. As stiffness of the connecting spring reduces further to $k = 0.085$, period doubling behaviour is seen and now amplitude of stress level fluctuates between two stages of vibration. However the stress build-up and drop mechanism is still in a periodic manner. Upon decreasing the stiffness of the sliding system to $k = 0.08437$, its behaviour goes to period quadrupling in which amplitude of stress fluctuates at four stages of vibration thus the slip surface is more prone to fail.  Finally the sliding system goes to chaotic behaviour at $k = 0.08421$ and now build-up and sudden-drop of stress amplitude is irregular and larger in comparison to the period doubling or quadrupling etc. thereby more chances of earthquake nucleation. These results are in confirmation with the Bifurcation diagram in Fig.7 of the manuscript.

---

## Author Comment (AC2) · 6 Mar 2016

We have provided the rebuttal to the reviewer's in the supplement.
* * *

---

## Referee Comment (RC2) · Anonymous Referee #2 · 20 Apr 2016

In this paper, the authors have analyzed the quasi-static frictional sliding of a spring-block slider system under the Rate-State Friction (RSF) formalism with three state variables. The study of 3-state RSF systems have not been carried out in the literature, the authors claim that this warrants the study of mechanical behavior of such systems. I feel that that this is not justifiable scientific motivation. There is good reason why 3-state variable systems have not been studied in the literature, laboratory friction experiments simply do not seem to suggest the need for this additional complexity. While some large rate-step experiments seem to point towards a second state variable (Marone, 1998; Ruina 1983) even this, in itself, is by no means a robust feature of experimental rate-stepping data. I find it difficult to understand the need for adding yet

another, by all accounts unnecessary, degree of freedom to the system.

Besides this clear motivational short-coming, the paper suffers from a lack of proper discussion of the background of the RSF formalism. For example, in Eq. 1 (which misses the summation sign on $\theta$i), the authors neglect to mention that they are using the Slip law for their simulations. Further, there is no discussion of why they choose the Slip law for their simulations. There is also no discussion of how these existing formulations of RSF fail to explain observed experimental data. There needs to be some discussion of how sensitive their results are to the choice of the state evolution law i.e. 3 Slip law state variables versus 3 Aging law state variables. One would expect the critical stiffness estimate to be insensitive to this choice (given Aging and Slip laws are asymptotically identical near steady state) but behavior under large perturbations from steady state is likely to be sensitive to this choice.

Also, given that a considerable amount of work has been done on evaluating the critical stiffness for the 1- and 2-state variable state evolution equations (Gu et al., 1984, Becker, 2000), the authors need to discuss explicitly how their estimate of the critical stiffness relates to the known expressions of the critical stiffness for these simpler formulations of state evolution. In a related point, the authors make the following claim in the abstract: "Linear stability analysis shows that critical stiffness, at which dynamical behaviour of the sliding system changes, increases with number of state variables"- it is likely that this conclusion is not generally true. This paper would definitely benefit from a section which systematically studies how the critical stiffness varies as a function of A, B1, B2, B3, L1, L2 and L3.

There are other technical problems with the paper:

Pg. 2, L6: It is incorrect that a second state variable was required to explain chaos in frictional slip. The second state variable is used to explain experimental observations of slip-weakening in response to rate steps.

Pg. 3, L4, L10: The conditions on the Lyapunov exponents for hyper-chaos as stated

are confusing. Does the sum of all LE's require to be +ve or -ve? The authors seem to suggest both at some point.

Pg. 12, L9: How is the fractal dimension 5.7 for a 4D (3 states, 1 slip rate) system?

The writing style, grammar needs to substantially improve throughout for this paper to be publishable in an international journal. Copy-editing by a native English speaker might be necessary.

Overall, given that the entire exercise carried out by the authors has very little scientific motivation based on laboratory friction experiments, I find it hard to recognize what exactly would the field of friction constitutive equations stand to gain from this work.

---

## Author Comment (AC3) · 15 May 2016

Dear Editor, We have replied/clarified the comments made by the second reviewer. We have uploaded the our reply to the comments made by the second reviewer. We have also uploaded the modified manuscript and indicated all changes made in the manuscript. Please write us if you need any clarification or correction from our side. Thanks, Arun

Please also note the supplement to this comment: http://www.nonlin-processes-geophys-discuss.net/npg-2016-11/npg-2016-11-AC3-supplement.pdf

[Figure]

[Figure]

**Supplement:**

**Dear Referee,**

Thanks for your constructive comments. We have tried our best to rebuttal your comments line by line as following:

(1)Referee: why 3-state variable systems have not been studied in the literature, laboratory friction experiments simply do not seem to suggest the need for this additional complexity. While some large rate-step experiments seem to point towards a second state variable (Marone, 1998; Ruina 1983) even this, in itself, is by no means a robust feature of experimental rate-stepping data. I find it difficult to understand the need for adding yet another, by all accounts unnecessary, degree of freedom to the system.

Authors: we agree with above opinion of the reviewer. But our Motivation to study the stability of 3sRSF emanates from the numerical simulations concerning the effect of temperature, viscosity as well as normal stress on chaotic behaviour of the 2sRSF model with the slip law. Our study establishes that either of these parameters (temperature, inertia, viscosity etc.) always results in diminishing the chaotic behaviour of the sliding system. However if an additional state variable " $\theta_3$ " is added in the 2sRSF model, unlike the effect of temperature, viscosity or normal stress on chaos, the present dynamic system that is,3sRSF model, in fact, becomes more chaotic. For instance, Fig.1 presents the results concerning the effect of temperature parameter "q" on chaos. As q varies from q = 1 (corresponds to the RSF model) to q = 1.01, period doubling oscillations reduces to periodic oscillations. It is important to mention that the Ruina-Rice slip law/slip law shows the chaos with 2sRSF model but the chaotic behaviour is not seen with the Dieterich-Ruina aging law/aging law. This is despite the fact that the both friction laws for stable variable " $\theta$ " were derived from the same expression for friction of rock surfaces (Ruina, 1983). The reason for this contradiction is not known in literature. So we believe that the present study will be useful to unravel the reason behind aforementioned "chaos contradiction". Although we have not included the results in Fig.1 in the present manuscript, the results (Fig.1) can be included in the manuscript if editor/reviewers suggest us to do so.

Fig.1 Effect of temperature parameter q on chaotic behaviour of the 2sRSF model with slip law in the form of phase diagram f vs.  $\phi$  for q = 1.0 and q = 1.01,  $\beta_1 = 1.0$ ,  $\beta_2 = 0.84$ ,  $\beta_3 = 0.38$ ,  $\rho = 0.048$  and  $\rho_1 = 0.034$  for initial condition [0,0,0,0]. It is obvious from two plots that period doubling reduces to single period orbits just upon slight modification in q.

(2)Referee: Besides this clear motivational short-coming, the paper suffers from a lack of proper discussion of the background of the RSF formalism. For example, in Eq. 1 (which misses the summation sign on  $\theta$ i), the authors neglect to mention that they are using the Slip law for their simulations. Further, there is no discussion of why they choose the Slip law for their simulations.

Authors: We have further modified the background of the RSF formalism. It is also clarified that the slip law of the RSF model is presently used in the present numerical simulations. We have also corrected the missing summation sign ( $\Sigma$ ) in state variable" $\theta$ ". It is already reported in literature that the 2sRSF with the slip law shows the chaos but the same is not seen with the aging law (Liu, 2007).

**We have also added in the manuscript at page no 2 and line no 18-25 as following:**

"Motivated from the numerical simulations concerning the effect of temperature, viscosity as well as normal stress on chaotic behaviour of the 2sRSF model with the slip law. We find that either of these parameters (temperature, viscosity etc.) results in diminishing the chaotic behaviour of the sliding system. However if an additional state variable " $\theta_3$ " is added in the 2sRSF model, unlike effect of temperature, viscosity or normal stress on chaos, the present dynamic system becomes more chaotic. Ruina(1983) has predicted that the RSF model with more than two state variables should also show the complex dynamical behaviour".

**(3)Referee: There is also no discussion of how these existing formulations of RSF fail to explain observed experimental data.**

Author: We would like to clarify that the present study is not at all motivated from the fact whether the established RSF models in literature fails to explain observed experimental data. As mentioned in the beginning that present study motivated to study the effect of "third state variable" that is, " $\theta_3$ " after concluding that temperature, pore pressure, normal stress, inertia of the sliding as well viscous damping eliminates the chaotic behaviour of the 2sRSF model with slip law. Another motivation for the present study why does not chaos is seen with aging law? In fact, our recent studies have shown that the RSF model with the aging law does not show the chaos at all irrespective of number of state variables added in the friction model. Ruina (1983) has also stated that the sliding system may become more complex upon addition of two or more number of state variables in the slip law. That is why we have chosen the slip law with three state variables.

(4)Referee: There needs to be some discussion of how sensitive their results are to the choice of the state evolution law i.e. 3 Slip law state variables versus 3 Aging law state variables. One would expect the critical stiffness estimate to be insensitive to this choice (given Aging and Slip laws are asymptotically identical near steady state) but behaviour under large perturbations from steady state is likely to be sensitive to this choice. Authors: The RSF model with the aging law does not the chaotic behaviour at all irrespective of number of state variables considered in the dynamical sliding system. Further we do not see chaos with the aging law even for large perturbations. This is the reason why the present study focuses on slip law which shows the chaos even after adding one more state variable. We are agree with the reviewer that critical stiffness is the same (Singh and Singh, 2016 published Geophysical Journal International).

(5)Referee: In a related point, the authors make the following claim in the abstract: "Linear stability analysis shows that critical stiffness, at which dynamical behaviour of the sliding system changes, increases with number of state variables"-it is likely that this conclusion is not generally true. This paper would definitely benefit from a section which systematically studies how the critical stiffness varies as a function of A, B1, B2, B3, L1, L2 and L3. Authors: We have checked the effect of state variables on critical stiffness and find critical stiffness increases in the present dynamical system. Fig.2 presents the effect of the friction parameter on the critical stiffness of the sliding system. It is obvious from the plots that critical stiffness of the sliding system varies linearly with each of the friction parameters  $\beta_2$ ,  $\beta_3$ ,  $\rho$  and  $\rho_1$ . Since the trend is linear in every case as obvious in Fig.2, as a result we have presented only one result that is,  $k_{cr}$  vs.  $\beta_1$ . However the editor/reviewers suggest us to do so, we can add these results in the manuscript as well.

Fig.2. Effect of friction parameter on critical stiffness  $k_{cr}$  for  $\beta_2$ ,  $\beta_3$ ,  $\rho$  and  $\rho_1$  for keeping remaining are to be fixed as  $\beta_1 = 1.2$ ,  $\beta_2 = 0.84$ ,  $\beta_3 = 0.38$ ,  $\rho = 0.048$ ,  $\rho_1 = 0.034$

(6)Referee: Pg. 2, L6: It is incorrect that a second state variable was required to explain chaos in frictional slip. The second state variable is used to explain experimental observations of slip-weakening in response to rate steps. Authors: We are fully agreed with the reviewer that the second state variable was added in the RSF model to explain the step velocity experiments (Ruina, 1980, 1983; Tullis, 1986). But this is only friction law of the RSF model which shows the chaos. Now present study establishes that even adding one more state variable that is, third state variable, also shows the chaos. This is most significant result of the present paper.

**(7)Referee: Pg. 3, L4, L10: The conditions on the Lyapunov exponents for hyper-chaos as stated are confusing. Does the sum of all LE's require to be +ve or -ve? The authors seem to suggest both at some point.**

**Authors:** We clarify that the present system is not hyper chaotic as we initially claimed in the paper. This is despite the fact that all the Lyapunov exponents are positive. In order to be a hyper chaotic system, the sum of all Lyapunov exponents must be negative. This condition not being met by the present dynamical system as far as the sum of Lyapunov exponents are concerned. We wish to modify at Page no 14 and line no 5-11 as following:

"The present analysis of the 3sRSF model shows in Fig.8 that the magnitude of LEs are  $LE_1=1.6358$ ,  $LE_2=0.0525$ ,  $LE_3=0.0662$ ,  $LE_4=0.0294$ . Since sum all LEs is not negative, as a result the present dynamical system is not hyper chaotic."

**5. Conclusions**

We have established numerically that the three state variables based RSF model with slip law is also chaotic. The route of chaos is established to be period doubling bifurcation. Moreover, linear stability shows that critical stiffness of spring-mass model increases with number of state variables in the RSF model. Moreover, all Lyapunov exponents are positive thus the 3sRSF is more chaotic than the 2sRSF model. Nonetheless the present system is not hyper chaotic as sum of all Lyapunov exponents are not negative.

**(8)Referee: Pg. 12, L9: How is the fractal dimension 5.7 for a 4D (3 states, 1 slip rate) system?**

Authors: We have already clarified and corrected this error in the uploaded rebuttal to the first reviewer. Moreover, calculation of the fractal dimension of the present dynamical system, which is equal to 5.7, is not correct at all. Since all the Lyapunov exponents are positive thus the formula for calculating fractal dimension may not be valid. As an error, we shall remove this estimation from the present article. We have also removed the following from the paper the following paragraph at Page no.12 and line no 3-9 "The relationship

between the Lyapunov exponents and fractal dimensions is established by Kaplan and Yorke (1979). They have proposed the Lyapunov or Kaplan-Yorke dimension  $D_{KY}$  which is given by the formula:

$$D_{KY} = D + \frac{1}{|h_D + 1|} \sum_{i=1}^{D} h_i \text{ Where } D \text{ is the largest integer for which } \sum_{i=1}^{D} h_i > 0. \text{ As a result, } D_{KY} \text{ is a result, } D_{KY} \text{$$

convenient geometrical measure of objects in phase space if Lyapunov exponents are known. The fractal dimension of the present dynamical system is calculated to be as 5.70."

**(9) Referee: The writing style, grammar needs to substantially improve throughout for this paper to be publishable in an international journal. Copy-editing by a native English speaker might be necessary.**

Author: We have revised the whole manuscript with the help of a native English speaker. We believe the present manuscript has improved greatly. We have revised the ABSTRACT at page no 1 and line no 8-15 as following:

**Abstract**

In this article, we study linear and non-linear stability of the three state variables based rate and state friction (3sRSF) model with spring-mass sliding system. The motivation emanates from the observation in the numerical simulations, unlike "state" variable, temperature, normal stress etc. diminishes the chaotic behaviour of the sliding system. Linear stability analysis shows that critical stiffness, at which dynamical behaviour of the sliding system changes, increases with number of state variables. The bifurcation diagram reveals that route of chaos is period doubling and this has also been confirmed with the Poincaré maps. Moreover, since all Lyapunov exponents of the present dynamical system are positive there by the present 3sRSF is not hyper chaotic. Finally, we believe that this study may be useful to establish the reason why slip law results in chaos but not the aging law of the rate and state friction model.

We have also checked the results/plots in the present manuscript; we do not find any error except slight improvement in Figs.6 & 8. Otherwise the results are alright. The modified plots are as following: